

# Gene flow relates to evolutionary divergence among populations at the range margin

Peter Kaňuch[1], Berrit Kiehl[2,3], Anna Cassel-Lundhagen[3],
Ane T. Laugen[3,4,5], Matthew Low[3] and Åsa Berggren[3]

[1] Institute of Forest Ecology, Slovak Academy of Sciences, Zvolen, Slovakia
[2] Department of Ecology and Genetics, Uppsala University, Uppsala, Sweden
[3] Department of Ecology, Swedish University of Agricultural Sciences, Uppsala, Sweden
[4] Bioeconomy Research Team, Novia University of Applied Sciences, Ekenäs, Finland
[5] Department of Natural Sciences, University of Agder, Kristiansand, Norway

## ABSTRACT

**Background:** Morphological differentiation between populations resulting from local adaptations to environmental conditions is likely to be more pronounced in populations with increasing genetic isolation. In a previous study a positive clinal variation in body size was observed in isolated Roesel's bush-cricket, *Metrioptera roeselii*, populations, but were absent from populations within a continuous distribution at the same latitudinal range. This observational study inferred that there was a phenotypic effect of gene flow on climate-induced selection in this species.
**Methods:** To disentangle genetic versus environmental drivers of population differences in morphology, we measured the size of four different body traits in wild-caught individuals from the two most distinct latitudinally-matched pairs of populations occurring at about 60°N latitude in northern Europe, characterised by either restricted or continuous gene flow, and corresponding individuals raised under laboratory conditions.
**Results:** Individuals that originated from the genetically isolated populations were always bigger (femur, pronotum and genital appendages) when compared to individuals from latitudinally-matched areas characterised by continuous gene flow between populations. The magnitude of this effect was similar for wild-caught and laboratory-reared individuals. We found that previously observed size cline variation in both male and female crickets was likely to be the result of local genetic adaptation rather than phenotypic plasticity.
**Conclusions:** This strongly suggests that restricted gene flow is of major importance for frequencies of alleles that participate in climate-induced selection acting to favour larger phenotypes in isolated populations towards colder latitudes.

## INTRODUCTION

Geographical patterns in phenotypic variation often reflect evolutionary and ecophysiological processes in response to different environmental conditions, such as temperature, light-dark cycles and precipitation. Intraspecific variation in body size of

Corresponding author
Åsa Berggren, asa.berggren@slu.se

ectotherms is a well-documented phenomenon (*Atkinson, Begon & Fitter, 1994*; *Whitman, 2008*; *Terribile et al., 2009*; *Fabian et al., 2015*) and this variation often manifests as positive or negative size clines (*Van Voorhies, 1996*; *Blanckenhorn & Demont, 2004*; *Whitman, 2008*). Body size variation may be determined by either physiological constraints (e.g. smaller body sizes due to lack of resources such as optimal heat energy or nutrients), by local selection pressures acting to favour some phenotypes (e.g. sex-specific selection on individual's size associated with reproductive success and season length) or by their mutual effect (*Blanckenhorn et al., 2006*; *Chown & Gaston, 2009*; *Kaňuch et al., 2015*). While local selection pressures promote morphological differentiation among populations (*Angilletta & Dunham, 2003*; *Bolnick & Nosil, 2007*), gene flow between populations can lead to so-called genetic swamping when alleles from one population spread through the other population genotype (*García-Ramos & Kirkpatrick, 1997*; *Lenormand, 2002*). New alleles or a sudden change of allelic frequencies in the population may then counteract local phenotypic adaptation (*Kawecki & Ebert, 2004*; *Crispo, 2008*; *Tigano & Friesen, 2016*; *Pedersen et al., 2017*). In contrast, reduced level of gene flow increases genotypic differentiation between populations while phenotypic divergence is often correlated with the degree of isolation (*Lenormand, 2002*). Therefore the relative effect of selection pressures depends on both the strength of the selection and on gene flow that is determined by the degree of isolation between populations (*Hendry, Taylor & McPhail, 2002*; *Raeymaekers et al., 2014*; *Berner & Thibert-Plante, 2015*). If gene flow among populations or sub-populations differs throughout a species' range of distribution, this gene flow will contribute to a varying degree to the phenotypic differentiation that is primarily driven by environmental factors (*García-Ramos & Kirkpatrick, 1997*).

Roesel's bush-cricket, *Metrioptera roeselii* (Orthoptera: Tettigoniidae), is widespread and continuously distributed across continental Europe (*Harz, 1957*; *De Jong & Kindvall, 1991*; *Hendry, Taylor & McPhail, 2002*) and has recently expanded its range northwards via several human-mediated long-distance colonisation events, which have been revealed by genetic traces (*Kaňuch, Berggren & Cassel-Lundhagen, 2013*; *Preuss et al., 2014*). At the northern range limit the species displays two different geographical distribution patterns. On the eastern side (the Baltic States and Finland) the distribution range extends as interconnected populations all the way to the polar circle (*Karjalainen, 2009*). On the western side (Denmark and the Scandinavian Peninsula) and some islands in the Baltic Sea (Åland Islands, Saaremaa), the species occurs in scattered local populations isolated from each other (*Albrecht, 1963*; *Ahlén, 1995*; *Bavnhøj, 1996*; *Karjalainen, 2009*; *Kaňuch, Berggren & Cassel-Lundhagen, 2013*). The species is characterised as a habitat generalist grassland-dwelling species (*Bellmann, 1985*; *Detzel, 1998*; *Ingrisch & Köhler, 1998*) where climate variation is the main source of selection pressure acting on local populations (*Kenyeres & Cservenka, 2014*). Wide ranges of morphological adaptations to climate have evolved in grassland insects (*Barnett & Facey, 2016*) and climate-determined season length also drives positive size clines in some Orthopteran species (reviewed by *Whitman (2008)*). While climate characteristics of *M. roeselii* populations in the two different geographical distribution patterns in northern Europe are similar across the extent of their distribution range

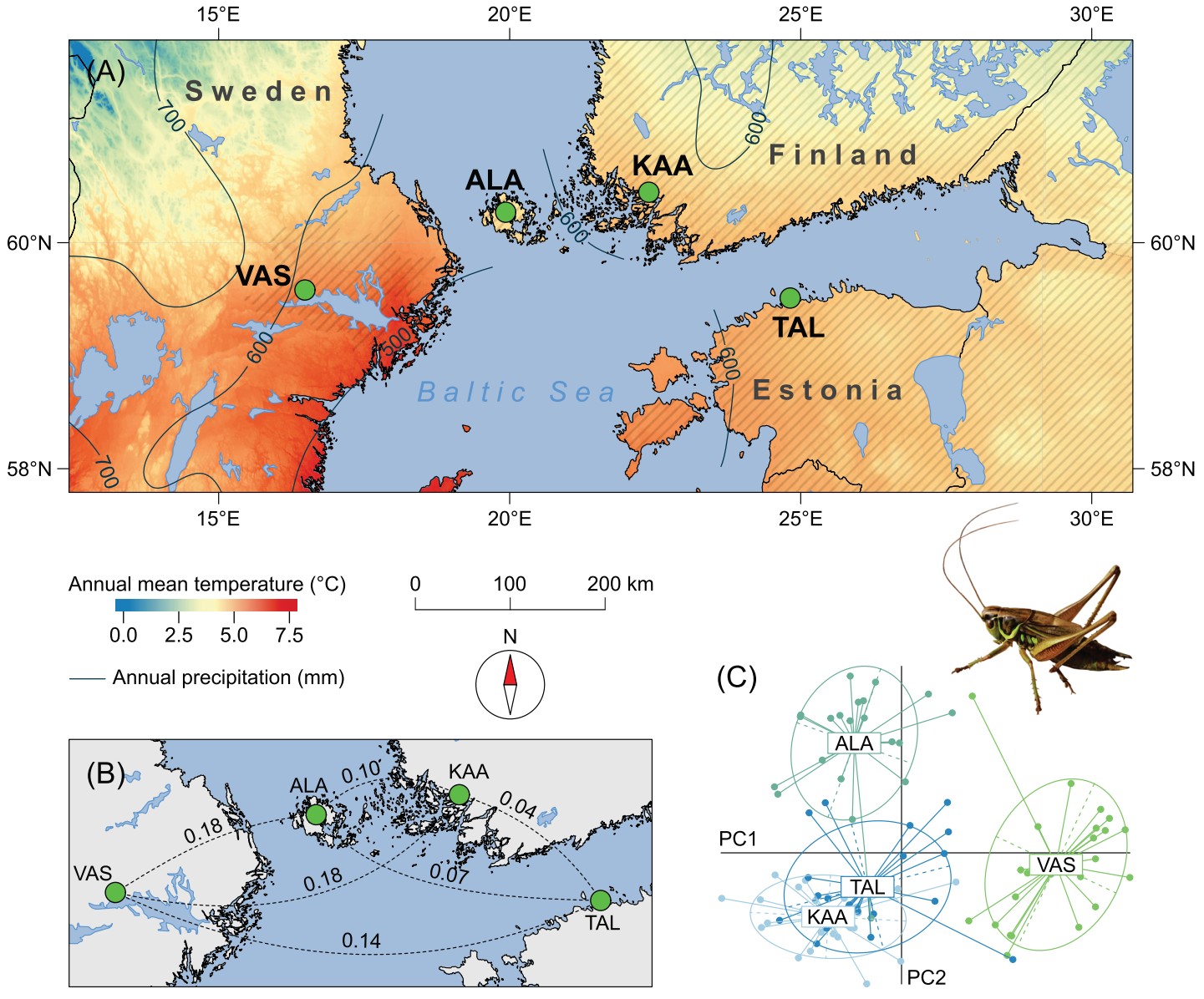

**Figure 1** **Study sites (circles) within the range of *Metrioptera roeselii* in northern Europe where nymphs and adults were sampled.** (A) The populations KAA and TAL are situated in the species continuous distribution range and connected via Russia in the east; VAS and ALA are isolated population sites. Hatched area shows range in 2010. For details about colonisation history see *Kaňuch, Berggren & Cassel-Lundhagen (2013)*. Climate data were downloaded from WorldClim.org database in ~1 km resolution. (B) Pairwise genetic differences between populations by $F_{ST}$ values corrected for null alleles by the ENA method. All values are significant according to a *G*-test (*P* < 0.05 after Bonferroni correction). (C) Genetic distances between individuals sampled in 2008 examined by a Principal Coordinates Analysis (PCoA) of the R-package 'adegenet' 2.1.1 (*Jombart, 2008*). Ellipses indicate credible distribution of the individuals in to different clusters.      Full-size  DOI: 10.7717/peerj.10036/fig-1

(*Cassel-Lundhagen et al., 2011*), the genetic distances between isolated populations and their inferred founders indicate a lack of ongoing gene flow into the isolated western populations. This is in contrast to the high level of gene flow among populations within the continuous eastern distribution range and restriction of gene flow resulted in contrasting patterns in both mitochondrial and microsatellite genetic structures of these populations (*Kaňuch, Berggren & Cassel-Lundhagen, 2013*). This peculiar distribution of
*M. roeselii* and its ecological characteristics, thus offers a unique opportunity to test the influence of variation in homogenising gene flow on phenotypic differentiation that can be associated with adaptive divergence in body size as an evolutionary response to selection in a cold climate environment.

By using a set of seven latitudinally-matched population pairs in which one population was geographically isolated and the other was part of the continuous distribution range, *Cassel-Lundhagen et al. (2011)* found that the latitudinal body size variation in *M. roeselii* within the two areas (continuous distribution vs. isolated populations) differed. Field collected adults from isolated populations were successively larger at higher latitudes (i.e. a positive size cline), while individuals from continuous populations remained similar in size in all locations for most of the measured traits. Along with that, significantly higher $F_{ST}$ values indicated no dispersal between isolated populations in contrast to the area of continuous species distribution. This observational study suggested that the size variation in the isolated populations was an adaptive response to local conditions at different latitudes, while the lack of a similar latitudinal response in continuous populations resulted from gene flow from surrounding areas preventing them reaching the same local adaptive optima. However, the contribution of phenotypic plasticity on the observed latitudinal differences in body size could not be definitively determined.

To investigate if the observed difference in body size between isolated and continuous-range populations of *M. roeselii* could be a genetically based adaptive response to selection or was simply phenotypic plasticity, we carried out a combined lab- and field-based experimental study. The size of four different body traits was measured in adult individuals of both sexes that originated from either laboratory or field conditions in a subset of seven previously studied population pairs. We controlled for random variation in the study populations' environments using individuals reared from early instars under laboratory conditions from both isolated and continuous populations, and compared their differences to adult individuals collected from the same populations in the wild. This allowed us to examine genotype-by-environment interactions in genetically distinct populations occurring at about 60°N latitude in northern Europe that were expected to show the most pronounced size differences in similar environment (*Cassel-Lundhagen et al., 2011*). If previously observed latitudinal variation in adult morphology was the result of local evolutionary adaptation rather than phenotypic plasticity, then the size differences between isolated and continuous populations originating from the same latitude should remain in the laboratory-reared individuals. If the size differences between the two populations was reduced or absent in laboratory-reared individuals, this would indicate a substantial phenotypic plastic component determining the variation between populations.

## METHODS

### The species

Roesel's bush cricket is a medium-sized orthopteran species with moderate sexual size dimorphism (the body length is 13–18 mm for males and 16–20 mm for females; *Harz, 1957*). In the field it is easily identified by the adult males' characteristic stridulation

that is audible almost continuously during the summer and early autumn days if the weather is warm and sunny (*Bellmann, 1985*). Through its wide distribution range in Europe, and recent western and northern distribution expansion (*Kaňuch, Berggren & Cassel-Lundhagen, 2013*), it inhabits varied grassland habitats where it feeds on plant matter and small insects (*Harz, 1957*; *Ingrisch & Köhler, 1998*; *Maas, Detzel & Staudt, 2002*). Mated females lay their eggs in hollow grass stems or other plant substrates and the nymphs hatch in spring, the first or second year after eggs are laid. The nymphs go through six or seven instars before they are fully developed and in northern Europe individuals usually reach maturity in July. Environmental factors as well as interactions with other individuals influence the species dispersal behaviour between habitats (*Berggren, 2004*, *2005*; *Eriksson, Low & Berggren, 2013*). The species has two wing forms; a common short-winged form that disperses mainly through walking and an easily identified rare long-winged form that is capable of active flight and usually occurs during rapid colonisation events. It has been experimentally shown that macropterism can be environmentally induced (*Poniatowski & Fartmann, 2009*), but the frequency of long-winged individuals in our study populations was very low (1.7%; *Kaňuch, Berggren & Cassel-Lundhagen, 2013*) and were not sampled in the field nor reared in the laboratory.

## Sampling

We selected four sampling sites that were located at ~60°N latitude close to the Baltic Sea coast in northern Europe with the most pronounced differences in individuals' body size but similar climate and environment (*Cassel-Lundhagen et al., 2011*; Fig. 1A), and that represent two pairs of genetically distinct populations ($F_{ST}$ = 0.04–0.18; Figs. 1B and 1C; populations were genotyped by seven polymorphic microsatellite loci in the study by *Kaňuch, Berggren & Cassel-Lundhagen (2013)*). Two sites represented the continuous range of the species distribution in Finland and Estonia (KAA, TAL) and two sites represented isolated populations on the Åland Islands and in Sweden (ALA, VAS). These sites contain well established populations and were sampled from the centre of their distribution at each site, in habitat patch size of ca. 2–4 ha, to avoid any possible issues with morphological differences arising from dispersing/colonising individuals at the range margin, or in establishing populations. In order to disentangle the effects of genetic versus environmental drivers of morphological differences in isolated populations of *M. roeselii*, we used individuals collected from three different sampling occasions at each of the four sites. The first sampling was of adult individuals collected in the field between 9 August 2008 and 10 September 2008 (from the previous study by *Cassel-Lundhagen et al. (2011)*). The second sampling was of first and second instar nymphs between 3 June 2010 and 9 June 2010 (approximately 60 nymphs per site) that were then reared under controlled laboratory conditions. The third sampling was adults collected between 26 August 2010 and 29 August 2010. Thus, we had wild-caught adults from all sites in different years (2008 and 2010) to examine between-year effects of different natural environmental conditions on morphological traits from isolated versus continuous populations. In addition, we had individuals sampled from all sites in the same year (2010) as nymphs and reared under controlled conditions to compare with wild-caught adults
**Table 1 Numbers of sampled *Metrioptera roeselii* adults.**

| Site (code) | Isolation level | Latitude (°N) | Longitude (°E) | Field grown (2008) | (2010) | Laboratory reared |
|---|---|---|---|---|---|---|
| Kaarina (KAA) | Continuous | 60.43 | 22.39 | 13/11 | 10/2 | 19/14 |
| Talin (TAL) | Continuous | 59.52 | 24.82 | 23/1 | 12/2 | 26/21 |
| Åland (ALA) | Isolated | 60.26 | 19.93 | 12/12 | 10/14 | 18/24 |
| Västerås (VAS) | Isolated | 59.59 | 16.48 | 12/12 | 6/8 | 28/23 |

Note:
Numbers of sampled *Metrioptera roeselii* adults (males/females) from sites representing continuous range of the species' distribution and sites from reproductively isolated populations in northern Europe.

from the same cohort exposed to natural environmental variation. In Orthoptera, initial offspring size is generally correlated with egg size, which is determined by the mother's size with some paternal contribution (*Weigensberg, Carriere & Roff, 1998*). Although temperature, moisture and photoperiod can generate variation in embryonic development of *M. roeselii*, especially in length of diapause (*Ingrisch, 1986a*, *1986b*, *1986c*), there is no evidence that other indirect maternal determinants of embryonic environment can significantly contribute to the variation in offspring body size of this or related species. As eggs of all tested populations have developed in very similar environmental conditions of semi-natural managed mesophile grassland habitats located close to sea level and at the same latitude, where annual mean temperature was about 5 °C and precipitation about 600 mm (Fig. 1A), we were confident with using early instars hatched in the field for our laboratory rearing. There was no indication that other environmental factors (e.g. different communities, competition or predation levels) could result in variation of local body size. To further minimise the risk of maternal effects confounding our results we ensured that habitats were sampled randomly, ensuring mixed origin of independent clutches. Using field-caught nymphs was also necessary because (similar to the study of *Simmons & Thomas (2004)*) we had 100% mortality in nymphs from captive-reared eggs.

The number of adults sampled from the wild in 2008 and 2010 ranged from 6 to 23 males and 1 to 14 females per site and year (Table 1). The nymphs collected in 2010 were after transport to the lab housed in individual cages (dimensions of 8 × 10 × 10 cm) with ad libitum access to food (fresh grass, pollen, fruit muesli and fortified dietary pellets Rep-Cal® Cricket Food) and water in a climate controlled room with natural and warm light in Uppsala, Sweden. All populations were thus kept in the same conditions with a temperature ranging from 23 to 25 °C, and because of natural lighting from north-facing windows the day-night cycle was kept the same as individuals in the field were exposed to. Such parameters contributed to the insects' physiological well-being that ensured easy nymphal development in the rearing facility (*Ingrisch, 1978*). The nymphs were moved to new clean cages every week to ensure optimal standardised rearing conditions. Despite some natural mortality during preimaginal development we reared 18–28 adult males and 14–24 adult females from each of the four sites to adulthood in the lab (Table 1).

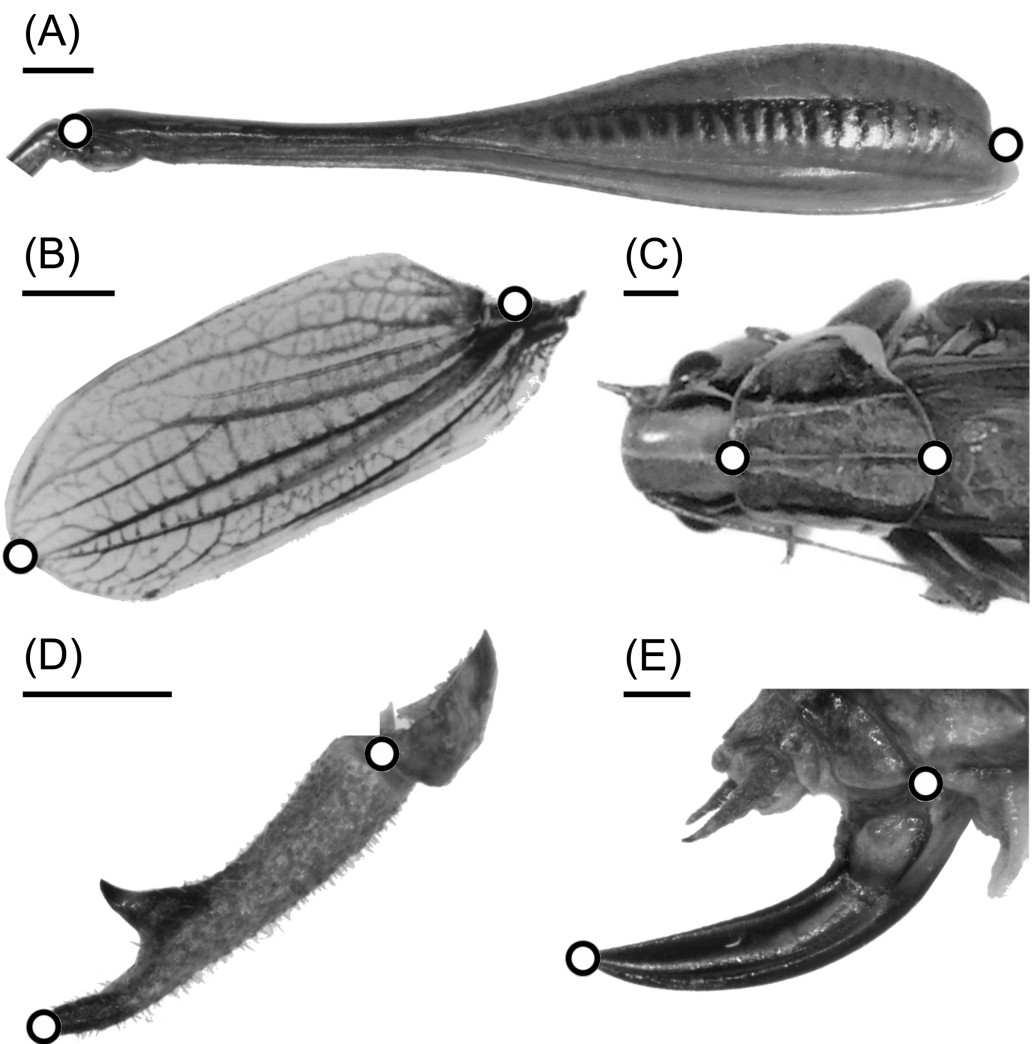

**Figure 2 The locations of landmarks (white circles) for size measurements of morphological traits of adult *Metrioptera roeselii* individuals.** (A) Hind femur, (B) forewing, (C) pronotum, (D) male's cerci and (E) female's ovipositor. The black line at each trait represents a scale bar of 1 mm.

After about three days after the final moult and reaching of adulthood, all individuals were stored in 95% ethanol and kept in room temperature until measured.

## Morphological measurements

Adult body size measurements were based around four morphological traits for each sex: the length of the femur, forewing, pronotum and genital appendages (male's cerci and female's ovipositor; Fig. 2). In paired organs, we used the right counterpart. These traits were chosen because they represent a range of responses to latitude found in *Cassel-Lundhagen et al. (2011)*; note that while femur, pronotum and gential appendages showed obvious differences in the size cline response between isolated and continuous populations, wings showed little or no size-cline difference (*Cassel-Lundhagen et al., 2011*).

For wild-caught adults in 2008, measurements were done using digital hand-held callipers. However, for the laboratory reared and wild-caught adult individuals sampled in 2010, measurements were undertaken using a different digital technique to reduce observer error in measurement. For this, the body parts of interest were digitally photographed at a standard focal distance and the distance between landmark points (Fig. 2) was measured using the software tpsDIG2 (*Rohlf, 2006*). To ensure the measurements taken using callipers in 2008 were comparable to those using the digital photography method in 2010 (individuals from the *Cassel-Lundhagen et al., 2011* study were not available for re-measurement), 48 individuals collected in 2010 (24 males and 24 females) were measured using both techniques for all traits. Using these data, a simple calibration model was then used to convert all calliper measurements to an equivalent digital measure for the 2008 data (Appendix S1 and S2). Although the two ways of measuring traits were strongly correlated (correlation coefficient $r = 0.85–0.99$) and allowed a high degree of precision when converting from manual to digital measurements, there was some uncertainty in the conversion factor. To account for this uncertainty, we used a Bayesian approach in both the calibration and subsequent analyses. This allowed us to use the uncertainty in the conversion estimate as a measure of observational error in the subsequent models (Appendix S1 and S2); this ensured that any error in conversion factors was accurately reflected in the errors of the estimates from the regression models described below.

## Statistical analyses

For each of the four morphological traits we fitted a general linear model that related the size of the trait to the following explanatory variables: (1) *isolation* to examine the difference between continuous and isolated populations, (2) *origin* for wild-caught versus laboratory-reared adults, (3) *sex* for males versus females (for femur, pronotum and wing) and (4) *year* to control for the between-year effect of different environmental conditions in the field in 2008 and 2010. Because of potential interactions between the first three variables, we also included the following second and third order interaction terms *isolation × sex, isolation × origin, origin × sex, isolation × origin × sex*. This allowed us to examine the effect of each of these factors and to produce model predictions for each trait based on sex, origin and degree of isolation. To account for any additional differences in morphology between sites, the site effect was included as an adjustment parameter on the intercept (a 'random effect'). To best estimate the range of probable values for each trait and to include the observational uncertainty from the calibration model (above), we used a Bayesian framework with minimally informative priors (see Appendix S1 and S2). The additional advantage of using a Bayesian approach is that all estimated variables and anything derived from these are posterior probability distributions. This allowed us to directly calculate the probability that traits in isolated populations were larger than those from continuous populations (either in general, sex-specific or origin-specific) by simple subtraction of the predictions for each group we were interested in (e.g. male lab-reared adult femur length from isolated population versus male lab-reared adult femur length from continuous population). Here, the

**Table 2 Model estimates of morphological trait length for *Metrioptera roeselii* adults.**

| Trait (sex) | Field grown individuals | | | Laboratory reared individuals | | |
|---|---|---|---|---|---|---|
| | Continuous | Isolated | $P_{\text{iso>con}}$ | Continuous | Isolated | $P_{\text{iso>con}}$ |
| Males | | | | | | |
| Femur | 13.02 ± 0.09 | 13.52 ± 0.08 | 1.000 | 12.51 ± 0.10 | 13.02 ± 0.09 | 0.999 |
| Wings | 8.86 ± 0.12 | 8.76 ± 0.11 | 0.261 | 9.01 ± 0.13 | 8.97 ± 0.12 | 0.404 |
| Pronotum | 4.06 ± 0.03 | 4.38 ± 0.03 | 1.000 | 3.88 ± 0.05 | 4.16 ± 0.04 | 1.000 |
| Cerci | 2.71 ± 0.03 | 2.97 ± 0.03 | 1.000 | 2.69 ± 0.04 | 2.96 ± 0.03 | 0.999 |
| Females | | | | | | |
| Femur | 13.97 ± 0.16 | 15.04 ± 0.09 | 1.000 | 13.56 ± 0.11 | 14.25 ± 0.09 | 1.000 |
| Wing | 6.20 ± 0.22 | 6.41 ± 0.12 | 0.795 | 6.20 ± 0.14 | 6.37 ± 0.14 | 0.856 |
| Pronotum | 4.40 ± 0.06 | 4.80 ± 0.04 | 1.000 | 3.98 ± 0.05 | 4.44 ± 0.04 | 1.000 |
| Ovipositor | 5.95 ± 0.06 | 6.40 ± 0.05 | 1.000 | 5.42 ± 0.06 | 6.09 ± 0.06 | 0.999 |

Note:
Model estimates of morphological trait length (means ± SD of the posterior distribution in mm) for *Metrioptera roeselii* adults categorised by sex, origin and genetic isolation. For each trait the probability that values are larger in isolated populations than in the continuous distribution range is given ($P_{\text{iso>con}}$).

proportion of the resulting posterior distribution that is above zero is the probability that group isolated > group continuous (shown in results as $P_{\text{iso>con}}$). Subsequent interpretation is that a probability of 0.50 indicates the mean estimate for the difference = 0 and has no predictive value; thus, parameters and derived variables where the posterior distribution has lower overlaps with zero can be considered increasingly important to the process being modelled (*Low et al., 2016*). We used a Bayesian Gibb's sampler (JAGS) called from R (*R Core Team, 2016*) using the 'rjags' package (*Plummer, Stukalov & Denwood, 2016*) to estimate final model parameters and generate predictions. For each model, we ran two independent chains and discarded the first 10000 values. Posteriors were estimated from 10,000 additional samples from the MCMC chain. Convergence was checked by visual inspection of trace plot stability and mixing. Model fitting was checked for the predicted means and coefficients of variation compared to the original data, using posterior predictive cheques based on 'Bayes $P$' values being between 0.1 and 0.9. We report posterior means and 95% credible intervals for estimated model parameters and predictions unless otherwise stated.

## RESULTS

Both males and females from the isolated populations had longer femurs, pronotums and genital appendages (cerci and ovipositor) than those from the continuous populations (Table 2; Fig. 3). For both sexes there was a high degree of certainty that these traits in the isolated populations were longer (probability > 0.99; Fig. 3; Tables 2 and 3), with this effect being consistent even when individual populations were considered separately (Table 3). Importantly, these patterns were independent of whether individuals were collected from the wild or reared under environmentally controlled conditions (Fig. 3, Tables 2–4). Thus, although crickets raised in the laboratory were generally smaller than their wild counterparts (Tables 2 and 4), the size differences between isolated versus

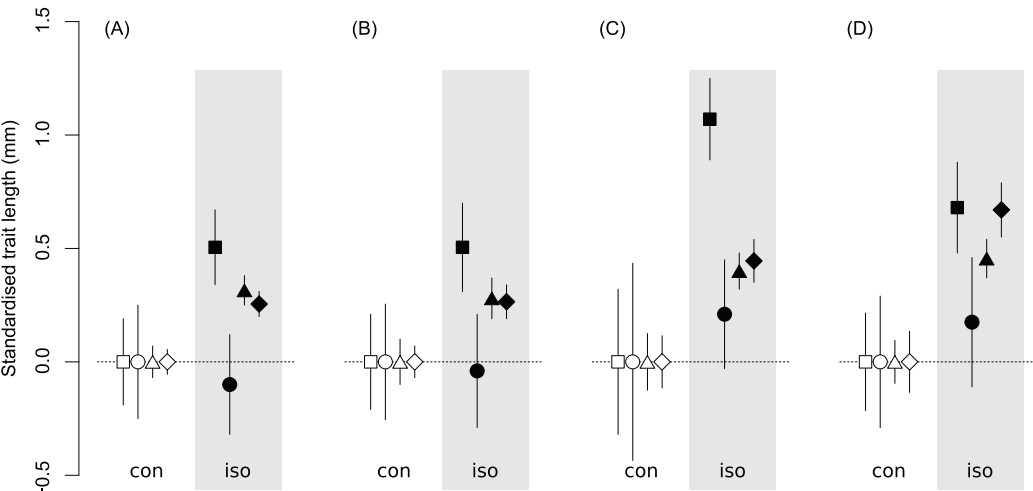

**Figure 3** The estimated medians and 95% credible intervals for the lengths of morphological traits of *Metrioptera roeselii*. (A) Males grown in the field or (B) reared in the laboratory and (C) females grown in the field or (D) reared in the laboratory. Squares, hind femur; circles, forewing; triangle, pronotum; diamond, cerci/ovipositor. Individuals originated from the sites of the continuous species range (con) and isolated sites (iso) located about 60°N latitude in northern Europe. Values on the *y*-axis are standardised relative to the estimates from the continuous populations, which are set to zero for each trait. For specific estimates of each trait, the differences between traits conditional on sex, rearing condition and genetic isolation, and the probability that they differ from each other see Tables 2–4.

**Table 3** Size difference (in mm) between traits measured in isolated versus continuous populations (where the difference is isolated—continuous), conditional on sex and rearing condition.

| Trait (sex) | Rearing conditions | | |
|---|---|---|---|
| | **Field** | **Laboratory** | **General** |
| Males | | | |
| Femur | 0.51 ± 0.12 | 0.51 ± 0.12 | 0.51 ± 0.09 |
| Wings | −0.10 ± 0.16 | −0.03 ± 0.15 | −0.07 ± 0.11 |
| Pronotum | 0.32 ± 0.04 | 0.28 ± 0.06 | 0.29 ± 0.04 |
| Cerci | 0.26 ± 0.03 | 0.27 ± 0.04 | 0.27 ± 0.04 |
| Females | | | |
| Femur | 1.06 ± 0.18 | 0.69 ± 0.12 | 0.87 ± 0.11 |
| Wing | 0.15 ± 0.23 | 0.17 ± 0.19 | 0.16 ± 0.15 |
| Pronotum | 0.39 ± 0.07 | 0.46 ± 0.03 | 0.42 ± 0.05 |
| Ovipositor | 0.45 ± 0.05 | 0.67 ± 0.06 | 0.56 ± 0.05 |

Note:
Rearing conditions are given for wild field-caught insects (field), those reared under environmentally controlled conditions (laboratory) and a general category where all observations are grouped regardless of rearing condition (general). Estimates are the means ± SD of the posterior distribution of the differences between populations, generated directly from the regression models (for details see Appendix S1).

continuous populations were similar for the laboratory-reared crickets when compared to wild crickets (Fig. 3; Table 3). For wings, however, there was no evidence that they were longer in isolated populations for males. The evidence for female wings being longer in

**Table 4 Site-specific trait estimates in mm (mean ± SD of the posterior distribution from the model) conditional on site isolation level where they were caught and where the insects were reared.**

| Site | Isolation level | Reared | Femur | Pronotum | Ovip./Cerci | Wing |
|---|---|---|---|---|---|---|
| Males | | | | | | |
| KAA | Continuous | Field | 13.02 ± 0.14 | 4.02 ± 0.06 | 2.74 ± 0.04 | 8.81 ± 0.19 |
| | | Laboratory | 12.21 ± 0.16 | 3.82 ± 0.07 | 2.67 ± 0.05 | 8.79 ± 0.20 |
| TAL | Continuous | Field | 13.01 ± 0.13 | 4.09 ± 0.04 | 2.70 ± 0.04 | 8.93 ± 0.16 |
| | | Laboratory | 12.71 ± 0.14 | 3.96 ± 0.07 | 2.71 ± 0.05 | 9.18 ± 0.16 |
| ALA | Isolated | Field | 13.79 ± 0.14 | 4.52 ± 0.05 | 2.95 ± 0.04 | 8.90 ± 0.19 |
| | | Laboratory | 13.02 ± 0.15 | 4.24 ± 0.07 | 2.89 ± 0.05 | 9.15 ± 0.20 |
| VAS | Isolated | Field | 13.39 ± 0.11 | 4.30 ± 0.04 | 2.98 ± 0.04 | 8.68 ± 0.14 |
| | | Laboratory | 13.01 ± 0.13 | 4.10 ± 0.05 | 3.03 ± 0.05 | 8.86 ± 0.15 |
| Females | | | | | | |
| KAA | Continuous | Field | 13.95 ± 0.19 | 4.40 ± 0.07 | 5.70 ± 0.09 | 6.19 ± 0.27 |
| | | Laboratory | 13.31 ± 0.16 | 3.86 ± 0.07 | 5.43 ± 0.11 | 6.02 ± 0.22 |
| TAL | Continuous | Field | 14.07 ± 0.32 | 4.40 ± 0.14 | 6.21 ± 0.09 | 6.16 ± 0.38 |
| | | Laboratory | 13.73 ± 0.14 | 4.04 ± 0.06 | 5.55 ± 0.09 | 6.39 ± 0.19 |
| ALA | Isolated | Field | 15.15 ± 0.12 | 4.88 ± 0.05 | 6.29 ± 0.05 | 6.81 ± 0.16 |
| | | Laboratory | 14.23 ± 0.14 | 4.43 ± 0.06 | 6.08 ± 0.09 | 6.42 ± 0.20 |
| VAS | Isolated | Field | 14.89 ± 0.14 | 4.72 ± 0.05 | 6.49 ± 0.06 | 5.88 ± 0.17 |
| | | Laboratory | 14.26 ± 0.14 | 4.43 ± 0.06 | 6.27 ± 0.08 | 6.37 ± 0.19 |

**Note:**
Consistent with the main results is that: (1) the longer trait length for isolated versus continuous populations in the wild is maintained when the insects are raised under controlled conditions, and (2) these results are consistent and show a high degree of certainty for the femur, pronotum and genital appendage (ovipositor/cerci) traits, and less consistent with a higher degree of uncertainty for the wing trait.

isolated populations was weak as the 95% CIs of the estimated difference between these groups overlapped zero (Fig. 3; Tables 2 and 3).

## DISCUSSION

Our laboratory findings suggest that phenotypic variation in both males and females of *M. roeselii* originating from genetically distinct populations in northern Europe is the result of local genetic adaptation rather than phenotypic plasticity. Reduced gene flow increases the probability of genotypic differentiation (*Endler, 1977*; *Lenormand, 2002*), but adaptive divergence in natural populations is the complex result of the balance between selection, gene flow and plasticity (*Hendry, Taylor & McPhail, 2002*; *Crispo, 2008*; *Raeymaekers et al., 2014*). By using controlled, though not fully 'common garden' rearing conditions we found no evidence that phenotypic plasticity contributed significantly to the observed size differences between isolated and continuous populations of *M. roeselii* because similar substantial size differences also remained in the laboratory reared individuals. Environmental settings could have a plastic effect on the size of some traits (at least in females); however, individuals that originated from genetically isolated populations (Figs. 1B and 1C) were always bigger (Fig. 3) when compared to individuals from latitudinally-matched areas characterised by continuous gene flow

between populations. Further, the fact that the founder sources of both isolated populations differed also indicates that possible genetic variation due to historic events does not seem to be responsible for observed morphological differences. ALA was most likely founded from the coast of the Gulf of Finland while VAS was founded from the Baltic coast in Poland with later introduction from Estonia (*Kaňuch, Berggren & Cassel-Lundhagen, 2013*), yet have similar morphological means and variances (*Cassel-Lundhagen et al., 2011*). Similarly, a clinal variation in the range of isolated populations (*Cassel-Lundhagen et al., 2011*) is probably not a result of genetic drift or founder effect, as all these populations have been colonised randomly from different sources (*Kaňuch, Berggren & Cassel-Lundhagen, 2013*). Thus although we cannot completely rule out genetic drift effects, the possibility that chance variation explains the clinal patterns of phenotypes found in isolated populations is highly unlikely. Variation in gene flow subsequent to establishment is therefore likely to be of major importance for phenotypic differentiation in this insect species. If such phenotypic variation is a result of an adaptive response to selection in isolated populations, this indicates that high levels of gene flow act to homogenise differences in the continuous populations (*Slatkin, 1987*; *Hendry, Taylor & McPhail, 2002*; *Raeymaekers et al., 2014*). Although it may appear that the KAA and TAL populations from the continuous species range are separated by the Gulf of Finland, a previous study has shown that there is intensive gene flow between them via their eastern land bridge in Russia (*Kaňuch, Berggren & Cassel-Lundhagen, 2013*). To better elucidate the spatial variation in selection between populations of different levels of gene flow, a comparison of $Q_{ST}/F_{ST}$ differentiations within and among groups would however be needed.

Despite general predictions on the disruptive effect of gene flow in adaptation, relatively little data exist on different effects of gene flow on local adaptation. Only recent development in genomics allow us to better understand how gene flow can promote adaptation via introgression of adaptive alleles through interbreeding of populations and how local adaptation can be maintained despite high level of gene flow due to spatial or temporal balancing selection (reviewed by *Tigano & Friesen (2016)*). Built on *Felsenstein's (1977)* work, *García-Ramos & Kirkpatrick (1997)* developed mathematical simulations of the interplay between gene flow and adaptation in peripheral populations that have restricted immigration of new individuals (and their alleles) from the main species' range. They demonstrated that a response to local selection pressures could result in a rapid and substantial evolution of traits when a population is isolated. Apparently, the relatively short time since the first introduction of Roesel's bush-crickets to the isolated sites studied (75–130 years; *Kaňuch, Berggren & Cassel-Lundhagen, 2013*), is enough time for the observed size variation to evolve. Our results are in agreement with larger body size in isolated populations from colder latitudes found in this species (*Cassel-Lundhagen et al., 2011*). However, rapid evolution of a positive Bergmann size cline is probably limited for small fast-developing insects. For them the short season in northern latitudes might not heavily reduce time for ontogenesis, contrary to that of large-sized

species with long development times (*Blanckenhorn & Demont, 2004*; *Whitman, 2008*; *Stillwell, 2010*; *Fabian et al., 2015*).

Genetic data have previously suggested that multiple introductions of individuals to the isolated populations have occurred since the first establishment of *M. roeselii*, but that there is little or no ongoing gene flow between these populations and surrounding or founding populations (see $F_{ST}$ values in Fig. 1B). Although some random genetic processes are still possible in generating the observed differences (*Kaňuch, Berggren & Cassel-Lundhagen, 2014*), climate-induced selection acting on the morphological traits appears to be strong enough to cause rapid divergence in a cold environment with a high level of seasonality (*Huey et al., 2000*; *Lawson & Weir, 2014*; *Krehenwinkel et al., 2016*). This does not seem to be the case, however, for wing length. In *Cassel-Lundhagen et al. (2011)* the size cline variation was similar for both isolated and continuous populations (i.e. positive but small in both males and females). This suggests that if any cline variation does occur for this trait, it may be driven by a factor that can overcome the homogenising gene flow effect in the continuous populations. However, given the results here and in *Cassel-Lundhagen et al. (2011)*, it is possible that this trait simply does not respond to latitudinal variation. The rate of trait evolution may also be elevated in novel environments during species' range expansion (*Krehenwinkel et al., 2016*). The hypothesis posed previously that the species adapt to local optima (*Cassel-Lundhagen et al., 2011*), is also supported by recent successful colonisation events in northern latitudes (*Preuss et al., 2014*).

Our study highlights that environmental factors do not exclusively drive the phenotypic expression of local adaptation (*Raeymaekers et al., 2014*). However, to learn about the adaptive effect of bigger body size and its expression in isolated populations of *M. roeselii* in northern Europe, an analysis of life history traits that maximise reproductive success is needed. Although very little is known about the environmentally determined effect of body size on reproductive behaviour, bush-cricket males have been found to prefer larger females in colder conditions (*Kaňuch et al., 2015*). Additionally, a female's size is a primary factor positively correlated to the number of offspring (*Honěk, 1993*), and body size is maternally inherited for both sexes (*Weigensberg, Carriere & Roff, 1998*). One concern regarding the interpretation of our study is the possibility of confounding maternal effects because reared nymphs were field collected and their parents and embryonic development was not controlled. While these effects must be acknowledged, we were careful to minimise their impact because individuals in the habitat were sampled randomly, ensuring mixed origin of independent clutches. In addition, nymphs collected were early instars whose age differential was <1 week, allowing us to be confident that maternal effects and/or different phenology contributed little to variation of nymphal development in the lab. If the species has a high rate of molecular evolution due to relatively quick generation turnover, high fecundity and short lifespan, restricted gene flow will probably not have deleterious effects in isolated populations (*Kaňuch, Berggren & Cassel-Lundhagen, 2014*). Thus to fully understand the effect of genetic drivers on species' phenotypic variation we need to also estimate the temporal and spatial extent of gene

flow associated with founder events (*Berggren, 2008*), and the possibility of maternal effects on phenotype by examining multiple generations of lab-reared individuals and their crosses. Such an approach could be possible in the case of *M. roeselii* in northern Europe due to known colonisation routes (*Kaňuch, Berggren & Cassel-Lundhagen, 2013*) and their ability to be reared in the laboratory that allow us to couple genetic data with morphological variation (*Cassel-Lundhagen et al., 2011*). In addition, a broader study utilising a 'common garden experiment' across the entire latitudinal range would remove any lingering doubts as to the relative role of genetics in these patterns and their possible interactions with latitudinal variables.

## CONCLUSIONS

Controlled laboratory conditions revealed no evidence that phenotypic plasticity contributed significantly to the observed differences in three selected body traits (lengths of femur, pronotum and genital appendages) between genetically distinct populations of *M. roeselii* in northern Europe. While climate characteristics of latitudinally-matched pairs of populations (Fig. 1A) are similar, lack of ongoing gene flow into the isolated populations (Figs. 1B and 1C) is implicated as a cause of these patterns. Thus, our results are consistent with expectations based on a tension between gene flow (as a homogenising force) and divergent climate-related selection (as a diversifying force) that plays out differently in isolated versus continuous populations.

## ACKNOWLEDGEMENTS

We thank Luc F. Bussière for comments on an earlier version of the manuscript and three anonymous reviewers are acknowledged for valuable suggestions which helped to improve our work.

### Funding

The study was supported by the Swedish University of Agricultural Sciences, the Slovak Scientific Grant Agency VEGA Grant number 2/0076/19 (Peter Kaňuch); and the Swedish Research Council Grant number VR 2012-03634 (Matthew Low). The funders had no role in study design, data collection and analysis, decision to publish, or preparation of the manuscript.

### Grant Disclosures

The following grant information was disclosed by the authors:
Swedish University of Agricultural Sciences.
Slovak Scientific Grant Agency VEGA: 2/0076/19.
Swedish Research Council: VR 2012-03634.

### Competing Interests

The authors declare that they have no competing interests.

## Author Contributions

- Peter Kaňuch conceived and designed the experiments, performed the experiments, prepared figures and/or tables, authored or reviewed drafts of the paper, and approved the final draft.
- Berrit Kiehl performed the experiments, authored or reviewed drafts of the paper, and approved the final draft.
- Anna Cassel-Lundhagen conceived and designed the experiments, authored or reviewed drafts of the paper, and approved the final draft.
- Ane T. Laugen conceived and designed the experiments, authored or reviewed drafts of the paper, and approved the final draft.
- Matthew Low analysed the data, prepared figures and/or tables, authored or reviewed drafts of the paper, and approved the final draft.
- Åsa Berggren conceived and designed the experiments, authored or reviewed drafts of the paper, and approved the final draft.

## Field Study Permissions

The following information was supplied relating to field study approvals (i.e. approving body and any reference numbers):

The study was on an insect and there are no regulations on the use of insects in scientific studies in Sweden (Swedish Animal Protection Ordinance 2019:66, chapter 7, § 6). We were very careful in handling and rearing of the animals. All applicable national and international guidelines for the care and use of animals were followed.

## Data Availability

Data and code are available in the Supplemental Files.

## Supplemental Information

Supplemental information for this article can be found online at http://dx.doi.org/10.7717/peerj.10036#supplemental-information.

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
