# Peer review of "Gene flow relates to evolutionary divergence among populations at the range margin"

_PeerJ, doi:10.7717/peerj.10036_

## Round 0.1 · original submission · Major Revisions

Dear Dr. Kaňuch and colleagues:

Thanks for submitting your manuscript to PeerJ. I have now received three independent reviews of your work, and as you will see, the reviewers raised some concerns about the research. Despite this, these reviewers are optimistic about your work and the potential impact it will have on research studying population biology and gene flow. Thus, I encourage you to revise your manuscript, accordingly, taking into account all of the concerns raised by both reviewers.

In your revision, please elaborate more on the assumptions of your approach and your interpretations of the data. As is, there is too little discussion supporting the manuscript’s findings.

Please ensure your figures are clear and present all of the relevant information.

While the concerns of the reviewers are relatively minor, this is a major revision to ensure that the original reviewers have a chance to evaluate your responses to their concerns. There are many suggestions, which I am sure will greatly improve your manuscript once addressed.

I look forward to seeing your revision, and thanks again for submitting your work to PeerJ.

Good luck with your revision,

-joe

Reviewer 1 ·

Basic reporting

Reporting is good in all respects - I did not check the raw data access.

Experimental design

This is original primary research, and the research question is well defined and meaningful. The study meets high technical and ethical standards (if licences were required, they were not reported in the ms). Methods were described with sufficient detail.

Validity of the findings

I believe the findings are interesting and conclusions follow from the data, with some caveats that could taken up in the discussion (see next box). The data is good, and the analyses well conducted. The findings and conclusions are by and large valid.

Additional comments

I think this was basically and interesting and valid study addressing a relevant problem. My main concern with the study design is that the lab reared nymphs were field collected, and hence, it is not possible to exclude the possibility that that carry-over environmental and/or maternal effects contributed to their development in the lab. Hence, the evidence for genetic basis of the observed differences is not bullet proof, but this was not discussed in the Discussion – I think it should. In the same vein, any differences in phenology among the sites – and thereby the age of field collected nymphs – might also play in. Some discussion on this would be also in place.

Some additional suggestions

L 27. I am not sure that “a phenotypic effect of gene flow on climate-induced selection in this species” was actually shown. The study did not measure selection, and the causality of phenotypic effect of gene flow was not established – it was inferred.

L84-85. I am not sure that it is accurate to say that climatic conditions are the same. In fact, no climate data is shown, but it is assumed that climate is the same because the latitudinal range from where populations were collected is the same. In fact, the Swedish locality and the Åland locality in particular experience a maritime climatic conditions, whereas the Kaarina population is more influenced by continental climate. Climate is not only about latitude. It would have been nice to see some more characterization of the climatic conditions in the study sites (e.g. growth season length).

L265-266. This basically ok, but as we know, even speciation is possible in the face gene flow. Gene flow is not necessarily a hinder for divergence. Perhaps this could be acknowledged somewhere.

L283-285. Agreed. But one way to elucidate this would be to calculate added variance components of population differentiation – what proportion of the variance is among vs within populations. It would give heuristic idea degree of population differentiation. As the authors have access to genetic data, they could also compare the FST with PST to gauge if the degree of divergence is likely to be driven by selection. I am not saying that they should do this, but I would have done because it might strengthen the case further.

L325-327 I had difficulties understanding this sentence.

L328-329. This feels like too strong statement – there is plenty of evidence that they do, and here it is not possible to rule out the effect of environmental factors as field collected nymphs were used.

L355-366. Maternal inheritance of size in crickets reinforces my concerns above – if maternal effects are strong, then field collected nymphs are perhaps not best way to infer genetically based divergence. More cautions interpretation – or discussion of potential caveats of the approach – would be in place.

Reviewer 2 ·

Basic reporting

The manuscript “Gene flow relates to evolutionary divergence among populations at the range margin” presents an interesting study about an aspect of dispersal ecology and evolution that is often discussed in theory or through models, but rarely addressed in real-world systems. The authors try to disentangle the effect of genetic structure and phenotypic plasticity on the phenotype of individuals in two different types of metapopulations: one sporting a low level of connectivity and one presenting regular gene flow.

Generally speaking, the manuscript is well-written, easy to follow and provides all the needed information even for a non-specialist reader to follow the narrative.
References are well-balanced, mixing classical seminal papers on the topic with more recent (< 10 years) articles on the subject.
Dataset structure is clear and linear.
The model applied has an easy structure (Appendix S1): the used priors are weakly informative to avoid constraining the MCMCs, however I would like to see a short explanation on how the priors were chosen (e.g. why those means and variances for the Normal distributions) to better follow the Authors reasoning.

The Authors declare that no specific field permit was needed to perform the experiment.


In general, I believe this paper is well-executed and suited for publication on PeerJ; I would only suggest a few, minor modifications to improve readability and address some issues I found while reading. Details are listed in the following fields

Experimental design

The project is well-structured and clearly exposed; the Introduction is detailed and provides a good overview of the state-of-the-art and the hypothesis the Authors want to test. Materials and Methods are quite well presented and adequate as well, even if I personally have some reserves about a few of the metrics that were used (see later).

Validity of the findings

Results are synthetic but nonetheless sufficient to prove the point the Authors wanted to show; in my opinion, clarity of the figures could be improved by performing minor modifications (see later).

Discussion is linear and clear, with the authors providing plausible explanations to their findings while at the same time recognizing the limits of their study without overestimating their work. I would have personally like to see a more propositive approach to some of the limits of the study: the details are addressed later on in this review.

Additional comments

Line-by-line breakdown of my comments:

- Line 49: “Interspecific variation in body size”. It could be useful to add here what some of the causes of this variation could be, not only listing its constrains and consequences.

- Line 64: I appreciate very much the digression about the consequences of gene flow. However, I feel that the topic is somehow incomplete: I would personally add a sentence about the “ideal” amount of gene flow and what happens when there is too little of it, in opposition to the “genetic swamping” already reported.

- Line 150 and following: I personally do not think that simply looking at the average yearly temperature is enough to state that the environmental conditions in two sites are similar. Many other factors could be involved in shaping the local conditions (e.g. humidity, exposition to sunlight, plant communities, presence of predators…), which in turn could affect individuals body size. As this point is pivotal for the whole experiment and discussion, I would like to see it discussed more in detail.

- Line 161: while I understand that no maternal effect has been ever spotted in the focal species, as the Authors explain later on in the paragraph, I am not convinced that collecting instars from the field and raising them in the lab would prevent any environmental effect to influence individuals growth. A more robust design would have been collecting females in the wild, let them lay eggs in the lab and then test those individuals: I would then kindly ask the Authors to provide a more robust backup for their claims.

- Line 188: were individuals stored in ethanol immediately after reaching adulthood or a certain amount of time was given between the last molt and the conservation?

- Line 205 and following: while I appreciate the effort the Authors put in being sure the results collected with two different methods are comparable, I can’t help but wonder why they changed sampling methods mid-way in the first place. A short sentence explaining their choice would be helpful here.

- Line 223: I believe that in the model should also be included the effect of the site the individuals were collected from. It is possible that one site produces inherently bigger or smaller individuals for any reason: with the model as-is, this would go undetected.

- Lines 243 and following: have the Authors performed other checks on the robustness and reliability on the model before proceeding? Standard Bayesian tests I am thinking about are the Rhat values and the Effective Sample Size of the final model.

- Line 283-85: to the Authors’ knowledge, are there any planned genetic studies to address the issue described here? If not, I think proposing a research line here (of course without going in detail) would help making the Discussion a more propositive and not simply a list of “things we do not know”.

- Figure 3: if the differences are calculated against the "Con" samples, those columns can be erased from the graph or placed in Supplementary if the Authors want to show the spotted confidence intervals.

Reviewer 3 ·

Basic reporting

The authors showed that R. roeslii from isolated populations have longer femurs, pronota and gentialia but shorter hindwings compared to continuous populations. They conclude that this is due to restricted gene flow and climate induced selection. Since laboratory-reared instars developed the same size as the individuals from the parental population, the authors ruled out the possibility that phenotypic plasticity based on environmental factors play a role in differences in body size.
The study is well written, interresting and overall easy to follow and understand. The provided context by the author, who shows great expertise in the background of natural history and distributional patterns of R. roeslii, was outlined very well. The figures are appropriate and easy to read and to understand. Fig 1 could profit from a northing and some additional data on genetics and population structure, which I outline in point 1 below. The authors provide the raw data inclusive the codes for analyses in a very useful manner, which I really appreciate.

Experimental design

The analyses is statistically sound. Sample size and number of populations is not huge, but sufficient for valid result with the used analyses.

I am not totally convinced that the first instars collected in the field could not have been influenced by environmental factors during embryonic development. Line 171 is not a good explanation und insufficiently cited. Since this is one of the aims of the study this could have been verified by taking adults from the field to the lab and breed them for at least another generation. This needs better references and explanation.

Validity of the findings

I generally agree with the conclusion.The different scenarios of how gene flow, genetic drift and adaptation can lead to different outcomes in isolated populations is well explained and referenced. However, the discussion on the casual mechanistic of genetics and drivers of adaptation could profit from a more straightforward explanation. Sometimes it is hard to differentiate between speculation and identification of causal drivers, which should be stated more clearly (but see point 2 below). Therefore, in my opinion, there are three general points, which I will outline in more detail, which need improvement before the manuscript would be ready for publication:
1. The title, abstract and general conclusion suggest that this is a genetic study. I was a little disappointed not to find any genetic data in this study. Maybe a rephrasing based on the actual results and methods used in this study would avoid disappointment by other readers. The previous study from the Author (Kanuch et al 2013) supports the conclusions based on genetic data. However it should be stated more clearly in the methods and discussion how exactly the sampled populations are structured genetically. Genetic structure and isolation could be easily depicted in Fig 1 instead of referring merely to another publication. That would save the reader a lot of work and increase readability, especially if the authors want to reach readers who are unfamiliar with the author’s former work. The study should be readable as a standalone work.
2. In line 118 you state: “If latitudinal variation in adult morphology is the result of local evolutionary adaptation rather than phenotypic plasticity, then the size differences between isolated and continuous populations should remain in the laboratory reared individuals.” In line 306 you discuss that the increase of body size is possibly due to a colder latitude. However, in the methods in line 173 you explain: “As eggs of all tested populations have developed in similar climatic conditions at the same latitude”. In line 353 you conclude that “characteristics of study populations are similar”. This would contradict line 40, 118 and discussion in line 306. I am confused: is it climate induced or not? When you discuss latitudinal or temperature affects or discuss “local selection pressures” (line 301-302) you should provide data on it (e.g. latitudinal and climate differences of the sampled populations).

---

## Round 0.2 · accepted · Accept

Dear Dr. Kaňuch and colleagues:

Thanks for revising your manuscript based on the concerns raised by the reviewers. I now believe that your manuscript is suitable for publication. Congratulations! I look forward to seeing this work in print, and I anticipate it being an important resource for groups studying population biology and gene flow. Thanks again for choosing PeerJ to publish such important work.

Best,

-joe

Reviewer 2 ·

Basic reporting

In general, the manuscript is of good quality; I refer to my previous review for any aspect not specifically addressed here.

During their revision, the authors managed to address the comments provided by the three reviewers, which often overlapped. They changed/expanded the methods section when possible to incorporate the comments and, when it was not possible, explained the rationale behind their choices.
The Discussion was also improved to address any doubt the Reviewers expressed.

Experimental design

No further comment; all the highlighted issues, both about the sampling and the analysis, had been addressed.

Validity of the findings

No further comment.

Reviewer 3 ·

Basic reporting

According to my first review, there were no concerns about the basic reporting. My recommendations to include genetic data in Fig 1 were met in a very pleasant manner.

Experimental design

The authors addressed the concerns I raised about the field collected nymphs and possible environmental factors. Under the described circumstances, although the method is not optimal (as explained captive reared individuals would be better, but is not possible), it is now addressed in a sufficient manner in the manuscript.

Validity of the findings

I was very pleased to see how the authors solved my comment about the inclusion of genetic results from previous studies. The new graphical illustration and the according explanations in the text improved the readability as a standalone work.

As explained in my previous review it was sometimes unclear and sounded even contradictory if the observed factors were climate induced or not. The authors rephrased the text and made their statement more clear.

I have nothing more to criticize and think the manuscript is fit for publication.